# Effects of Adjuvants on Spraying Characteristics and Control Efficacy in Unmanned Aerial Application

**Shilin Wang** [1,2]**, Xue Li** [1]**, Aijun Zeng** [2]**, Jianli Song** [2]**, Tao Xu** [1]**, Xiaolan Lv** [1] **and Xiongkui He** [2,*]

1   Institute of Agricultural Facilities and Equipment, Jiangsu Academy of Agricultural Sciences, Nanjing 210014, China; shilin@jaas.ac.cn (S.W.); lixue@jaas.ac.cn (X.L.); xutao@jaas.ac.cn (T.X.); 20140002@jaas.ac.cn (X.L.)
2   Center for Chemicals Application Technology (CCAT), College of Science, China Agricultural University, Beijing 100193, China; aijunz@cau.edu.cn (A.Z.); songjianli@cau.edu.cn (J.S.)
*   Correspondence: xiongkui@cau.edu.cn

**Abstract:** Pesticide application by unmanned agricultural aerial vehicles (UAVs) has rapidly developed in China and other Asian counties. Currently, tank-mix spray adjuvants are usually added into pesticide solutions to reduce spray drift and facilitate droplet deposition and control efficacy. The currently used tank-mix adjuvants are all derived from conventional ground sprays, and their mechanisms of action in aerial applications are still unclear. In order to clarify the spraying characteristics and control efficacy of those adjuvants in aerial sprays, the performances of various types of tank-mix adjuvants were compared by analyzing droplet spectrum, drift potential index (DIX) in a wind tunnel, field deposition and control efficacy on wheat rust and aphids. The atomization results showed that the addition of adjuvants could change the droplet spectrum of liquid, and the results suggest that droplet size is an effective indicator of spray drift potential. In the field application, the meteorological conditions are complex and uncontrollable, and the effects of adjuvants on droplet deposition and distribution were not significant. Compared with the control solution, there was no significant difference in the deposition amount of each adjuvant solution, and the CVs of deposition were higher than 30%. Adding adjuvants to the spray solution can significantly improve the control efficacy of pesticides on wheat aphids and rust and also prolong the duration of the pesticide. Our results suggest that tank-mix adjuvants should be added when UAVs are used for aerial application. This study can be used as a reference to the research and development or selection of adjuvants in aerial sprays of UAVs.

**Keywords:** unmanned aerial vehicle sprayer; pesticide application; tank-mix adjuvant; spray drift; control efficacy





## 1. Introduction

In Asia (China, Japan, South Korea, etc.), the average arable farm area is small, and the terrain is mountainous. It is difficult for ground-based plant protection machinery to apply pesticides in limited-access areas such as rice paddies and hillsides [1]. With the labor population migrating from rural to urban and the aggravation of population aging, there is an urgent need for new equipment for pesticides application that can adapt to small plots in hilly and mountainous areas [2]. In recent years, pesticide application by UAVs has rapidly developed in China and other Asian counties due to their suitability to complex terrain, high working efficiency, lower spray volume, reduction in labor intensity and pesticide contamination of operators [3–7].

Over the past few years, extensive research has been done about flight platforms and spraying systems of UAVs. Huang et al. [4] developed a spraying system for the UAV platform. The developed system could provide accurate and site-specific crop management when coupled with UAVs. In addition, a pulse width modulation (PWM) variable spraying system based on a miniature UAV was developed [8] that consisted of an airborne spraying

device and a ground control unit. The airborne spraying device was remotely controlled by the ground unit through a wireless data transmission module to achieve variable spray. Wang et al. [9] designed a bipolar contact electrostatic spraying system for UAVs; its charged droplets can produce a wrap-around effect on the underside of the leaves, which promotes droplet adhesion.

In terms of UAV spraying performance, extensive studies on droplet distribution and the control effect of pesticides have been conducted. In early studies, researchers found problems, such as uneven droplet distribution, serious drift potential and tube or nozzle blockage during UAV pesticide application [6,10–12]. The flying and spraying parameters of UAVs significantly influence droplet deposition and drift [13]. In order to optimize the deposition and drift characteristics of pesticide droplets of unmanned aerial spraying, researchers have conducted a lot of research on the application parameters of UAVs. Qiu et al. [14] studied the spraying performance of CD-10 UAVs under the influence of flight height and velocity. A relevant model was established to clarify the relationship between deposition concentration, deposition uniformity, flight height and velocity. Qin et al. [15,16] studied the influence of spraying parameters of N-3 UAVs on droplet deposition on maize canopies and control effects of insecticides sprayed with a UAV against planthoppers, respectively. Wang et al. [17] studied the relationship between the spray volumes of UAV and the deposition and control efficacy of droplets. As a result, the optimal application volume was determined, and the control efficacy was optimized. Li et al. [18] characterized the comparative differences of UAV technology and a conventional airblast sprayer and presented promising data to support crop protection programs for large canopy crops of unmanned aerial applications.

These studies have laid a solid foundation for unmanned aerial spraying, and the droplet distribution and deposition rate of UAVs have been significantly improved. At the same time, after years of practice and experience, avoiding the use of solid formulations, such as wettable powder (WP) or water dispersible granule (WG), could effectively reduce tube and nozzle blockages of aerial spraying system. Therefore, the types and numbers of plant protection UAVs have shown geometric growth, and the area of operations by UAVs has also increased significantly. At present, UAVs are widely used for pest and disease control in east Asia [1,19]. However, aerial application with UAVs still suffers from serious drifting behavior, which poses great safety hazards to the surrounding environment and sensitive crops. Nozzle type, nozzle size, formulation type and tank additives are commercially available drift reduction technologies designed to decrease drift through modification of the droplet size distribution upon atomization. However, most studies have only evaluated spray nozzles in wind tunnels and did not include the effect of tank mixes [20]. Currently, tank-mix spray adjuvants are usually added into pesticide solutions to reduce spray drift and facilitate droplet expansion and deposition [21]. Adjuvants are tank additives that are marketed for their enhancement benefits according to the function they are designed to perform. Some adjuvants are designed to enhance the performance of the pesticide, usually through better absorption, whereas others are designed to enhance the qualities of the spray by modifying the physical properties of the spray solution [22,23].

The process of pesticide application involves the fundamental phenomena of atomization, precipitation and deposition. Selecting appropriate adjuvants can improve the application quality by changing the atomized droplet spectrum, reducing the drift of precipitation and increasing the amount of deposition. However, the currently used tank-mix adjuvants are all derived from conventional ground sprays, and their mechanism of action in aerial applications is still unclear. In order to clarify the action mechanism of those adjuvants in aerial sprays, the performances of various types of adjuvants were compared by analyzing the whole process of aerial application, including droplet spectrum, drift potential, field deposition and control efficacy.

## 2. Materials and Methods

### 2.1. Unmanned Aerial Vehicle and Spray System

An oil-powered single-rotor UAV (3WQF120-12, Anyang Quanfeng Biological Technology Co., Ltd., Anyang, China) was selected for the experiment. The fuselage length, height and rotor diameter of the UAV (Figure 1) were 2130 mm, 670 mm and 2410 mm. The net weight of the UAV was 30 kg, the flight speed was 0–15 m/s and the flight height was 1–4 m. The spraying system of the UAV consists of a spray controller, tank, liquid pump, pipeline, spray nozzle and boom. The tank was connected to the filter, liquid pump, spray controller and nozzles in turn through the pipeline, and the nozzles were fixed on the spray boom. The tank had a capacity of 12 L, and the length of boom was 1250 mm. The boom was fixed 0.5 m below the rotor on the landing gear. Flat fan nozzles LU120-01 (Lechler GmbH, Metzingen, Germany) were installed at both ends of the boom, and the interval between the two nozzles was 1100 mm.

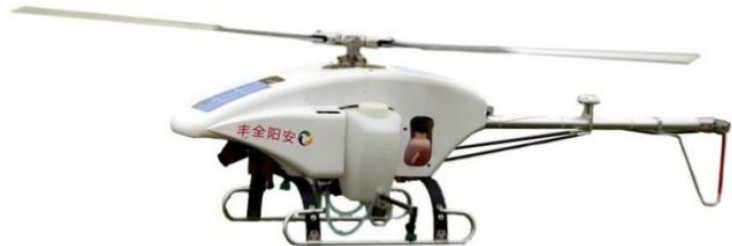

**Figure 1.** 3WQF120-12 plant protection UAV.

### 2.2. Adjuvants

The experiment selected a variety of tank-mixed adjuvants on the market for aerial application. The selected adjuvants were AS 100, AS A + B, Breakthru, Momentive, ND 500, ND 700 and QF-LY. The key commercial information and the physicochemical properties of those adjuvants at recommended concentrations are shown in Table 1. Compared with pure water, the viscosity of each kind of adjuvant solution does not change much at the recommended concentration, while the surface tension is significantly reduced.

**Table 1.** Commercial information, recommended concentrations and physicochemical properties of spray adjuvants.

| Name of Adjuvants | Main Ingredients | Manufacturer | Recommended Concentration/% | Viscosity (mPa·s) | Surface Tension (mN/m) |
|---|---|---|---|---|---|
| AS 100 | Methylated seed oil | Aishang, China | 0.5 | 1.36 | 29.44 |
| AS A + B | Methylated seed oil, organosilicon | Aishang, China | 1.0 | 1.49 | 30.67 |
| Breakthru | Mixture of fatty acid ester | Evonik, Germany | 0.5 | 1.24 | 31.20 |
| Momentive | Polymeric Components, organosilicon | Momentive, USA | 0.5 | 1.41 | 22.99 |
| ND500 | Hyperbranched polymer | Nuonong, China | 0.5 | 1.37 | 25.72 |
| ND700 | Hyperbranched polymer | Nuonong, China | 0.5 | 1.36 | 29.78 |
| QF-LY | Organosilicon | Quanfeng, China | 0.5 | 1.43 | 20.80 |

### 2.3. Experimental Design

#### 2.3.1. Droplet Spectrum

Droplet spectra of adjuvants at recommended concentrations and tap water were measured with a laser particle size analyzer (Mastersizer 2000, Malvern Panalytical Ltd., Malvern, UK) by scanning the complete cross-sectional area of the spray at the distance of 50 cm below the nozzle. The flat fan nozzle LU120-01 mounted on the indoor spray system was used to spray with a pressure of 0.3 MPa. Three replicate measurements were performed for each spray solution. The data were analyzed to determine the volume median diameter (*VMD*) and particle relative span (*RS*) of the spray. The *RS* represents

the uniformity of the atomized droplets, and the smaller the *RS* is, the more uniform the atomization is. The *RS* was calculated with Equation (1).

$$RS = \frac{DV90 - DV10}{VMD} \tag{1}$$

In the equation, $DV10$ is the diameter at which 10% of the droplet volume is contained in the droplets at or below this diameter. In the same way, the $VMD$ is 50%, and $DV90$ is the diameter at a droplet volume of 90%.

### 2.3.2. Drift Potential in the Wind Tunnel

An open-jet low-speed wind tunnel was used for drift potential measurements, which was 7.5 m in length, 1 m in width and 1 m in height. The length of the wind speed working section was 4 m, and the wind speed ranged from 0 to 8 m/s. The meteorological conditions were measured by Testo 350-454 (Testo GmbH) at 9 positions in a plane perpendicular to the airstream 2 m downwind from the nozzle, according to ISO 22856. The duration of the measurement at each position should be tested for no less than 10 s. The specific conditions during the test were as follows: nominal airspeed 2.01 m/s, temperature 12.3–14.8 °C and relative humidity 46%–57%.

In order to eliminate the impact of droplet splash from the ground, a virtual floor was protocolled at the height of 5 cm. The height of the nozzle from the ground was $h_N$ = 0.85 m (0.80 m from the virtual ground), and the spray fan was vertical to the crosswind direction. The measurement of drift was divided into sediment drift and airborne drift, and the test-specific layout of samples is shown in Figure 2. Sediment drift was measured by coated papers (5 cm × 9 cm) mounted on the virtual floor at 1.0,1.5, 2.0, 2.5, 3.0 and 3.5 m downwind from the nozzle, and two pieces of paper were placed at the same distance. The airborne drift was determined by 9 polytetrafluoroethylene (PETF) lines with an interval of 0.10 m on a vertical plane at 2 m downwind from the nozzle, in this case, the lowest line was 5 cm (equal to the virtual floor), and the highest line was 85 cm (equal to the nozzle height).

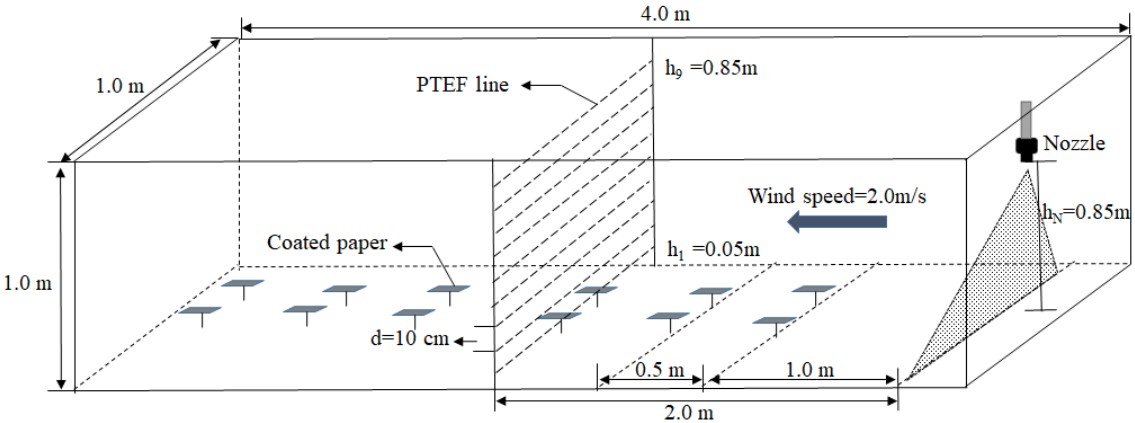

**Figure 2.** Layout of sediment drift and airborne drift samples in wind tunnel.

The flat fan nozzle LU120-01 was used to spray adjuvant solutions and tap water with a pressure of 0.3 MPa for 7.0 s, and each measurement was tested for three replications. Fluorescence tracer BSF (brilliant sulfoflavin dye, Chroma-Gesellschaft Schmid, Kongen, Germany), with a mass fraction of 0.1%, was added into the spray solutions for quantification. The dried coated papers and PTEF lines were removed from the wind tunnel after application then kept in the dark and low temperatures, respectively.

### 2.3.3. Field Test

Field trials were conducted from 2016 to 2018 in Neihuang County, Anyang City, Henan Province (2016 and 2018) and the Xinxiang Experimental Base of the Chinese Academy of Agricultural Sciences (CAAS) (2017) during the flowering period of wheat to compare the performance of adjuvants in aerial application. In the annual experiment, 9 plots of 40 m × 70 m were divided, of which 7 plots were sprayed with the added adjuvants, one plot was sprayed without adjuvants as a control and another plot was used to assess the disease and pest severity without any application. A buffer zone with a width of 20 m was set up between each treatment plot to avoid errors caused by spray drift. The control efficacy of wheat rust and aphids were evaluated based upon the actual occurrence of pests and diseases in the experiment field.

The optimal operating parameters of the 3WQF120-12 UAV were determined based on a large number of preliminary tests and application performance. The flight altitude of the UAV was 1.5 m above the wheat canopy, and the spraying swath was 4.5 m. The application volume of the UAV was 12 L/ha. The pesticides used in the test were 20% imidacloprid SL (ai.60 g/ha) and 40% tebuconazole SC (ai.480 g/ha). The recommended concentration of each adjuvant in spray solution was given in Table 1, and BSF fluorescence with a mass fraction of 0.2% was also blended into the solution as a tracer to quantify the deposited droplets. To collect the droplets, three sampling rows of filter papers ($\varphi$ = 70 mm) were distributed horizontally with the same height of the wheat plant in the central region of the test plots. There were 5 pieces of filter paper in each sampling row, and the interval between each paper was 1.0 m. After application, all the sampling filter papers were collected timely and placed into labeled Ziplock bags, then stored in the dark at $-20$ °C until analysis. All of the samplings were analyzed within 48 h in the laboratory. A quantitative volume of deionized water was used to elute the filter paper, and a fluorescence spectrometer (LS55, PerkinElmer Instruments, Waltham, MA, USA) was used to determine the amount of droplet deposition.

### 2.4. Sample Processing and Calculation

### 2.4.1. Deposition on the Samples

The coated paper, PTEF line and filter paper samples were washed with 100 mL deionized water for 1 min in sealed bags. LS55 fluorescence spectrophotometer was used to measure the fluorescence value of sample eluent at 501 nm when the excitation wavelength was 465 nm. In addition, the fluorescence value of each original spray liquid diluted 1000 times was also measured to calculate the volume of the liquid on samples.

$$Vs = \frac{V_w \times FL_S}{N \times FL_a} \times 10^3 \tag{2}$$

where $Vs$ is deposition on the sample, μL; $V_w$ is the volume of eluent, mL; $FL_S$, fluorescence value of eluent; $FL_a$, fluorescence value of diluted liquid; $N$, dilution times of spray liquid.

### 2.4.2. Drift Potential Index

The drift potential index (DIX) was calculated in order to estimate the drift reduction performance of the investigated adjuvants. The determination of spray drift potential includes both sediment drift and airborne drift. The DIX method considered not only the total deposit volume and the weight of the collection height or distance, but also the relationship between the data from the wind tunnel test and those from the field test by introducing two parameters [24,25]. For airborne drift, a vertical drift potential profile was calculated from the data by integration over horizontal measuring lines. The DIXs of spraying liquids were obtained as follows:

$$T = \int_0^{h_N} \int_0^\infty v(y, z) dy dz \tag{3}$$

$$V = T/T_N \tag{4}$$

$$h = \frac{\int_0^{h_N} V(z) \cdot z dz}{\int_0^{h_N} V(z) dz} \tag{5}$$

$$DIX = \frac{h^a V^b}{h_{St}^a V_{St}^b} \times 100\% \tag{6}$$

where $V$ is the volume flux at any point of the measuring plane, $T$ is the total volume flux over the drifting spray profile, $T_N$ is nozzle output, $V$ is the relative drift potential volume, $h$ is the characteristic height of the drift potential cloud, $h_{St}$ is the characteristic height of the drift potential cloud of the reference liquid (without adjuvants), and $V_{St}$ is the relative drift potential volume of the reference liquid. In Equation (6), parameters $a$ and $b$ were known from a regression analysis with wind tunnel and field measurement for a lot of nozzles, and the best fit was $a = 0.88$ and $b = 0.778$, respectively [24]. The characteristic distances and *DIXs* of sediment drift were calculated in the same way.

2.4.3. Control Efficacy of Pesticides

The survey and recording of wheat rust and aphids were carried out according to pesticide field efficacy test criteria. Based on the incidence of the blank control (untreated plot), three assessments of wheat rust infections were carried out during the tests. The first assessment was carried out prior to the pesticide's application. The second and third assessments were carried out at 7 and 14 days after treatment. Assessment was made by sampling five locations on each plot for the diseases. Each plot had 50 plants, and the flag leaf and the first two leaves under the flag of each plant were assessed. This was done according to the national standard of cereal rust severity classification, and the diseased leaf was assessed based on the diseased leaf area.

$$DI = \left( \sum \frac{N_d \times V_g}{N_t \times V_h} \right) \times 100\% \tag{7}$$

where $DI$ is the disease index, %; $N_d$ is the number of diseased leaves at each grade; $V_g$ is the representative value of the specific grade; $N_t$ is the total number of investigated leaves; $V_h$ is the representative value of the highest grade.

$$CE = \frac{DI_c - DI_s}{DI_{in}} \times 100\% \tag{8}$$

where $CE$ is the control efficacy of rust, %; $DI_c$ is the disease index in the control group; $DI_s$ is the disease index in spraying groups; $DI_{in}$ is the disease index increment in the control group.

The numbers of live wheat aphids were calculated before pesticide application and on 1, 3, 7 and 14 days after spraying in all treated plots and the blank control plot (untreated), ignoring the types or instars. Assessment of aphids was made by sampling five locations on each plot and counting the total number of aphids in 20 wheat plants of each location. The insecticidal dropping rate and correction control efficacy were calculated according to the following equations:

$$DR = \frac{BS - AS}{BS} \times 100\% \tag{9}$$

where $DR$ is the decrease rate of aphids; $BS$ is the number of live aphids before spraying and $AS$ is the number of live aphids after spraying.

$$CE = \frac{DR_T - DR_C}{100\% - DR_C} \times 100\% \tag{10}$$

where *CE* is the correction control effect of aphids; $DR_T$ and $DR_C$ are the decrease rate of aphids in treated plots and untreated plots, respectively.

## 3. Results

### 3.1. Effect of Droplet Spectrum

Table 2 shows the *DV*10, *VMD*, *DV*90 and relative span of droplet atomized at 0.3 MPa with the LU120-01 nozzle. For the control spraying solution, the *DV*10, *VMD* and *DV*90 were 76.1 μm, 151.6 μm and 272.4 μm, respectively, and the droplet relative span was 1.29. When AS100 was added to the spray solution, the *DV*10 and *VMD* of the droplets were 68.7 μm and 140.9 μm, respectively, which were significantly lower than that of the control solution; however, the *DV*90 was 264.5 μm, which had no significant difference with control solution ($p < 0.05$). For ND500 and ND700 solutions, their corresponding *DV*10, *VMD* and *DV*90 were all significantly lower than the results of the control solution. Although the adjuvants AS100, ND500 and ND700 changed the particle sizes of atomization, their relative droplet spans were not significantly different from that of the control solution ($p < 0.05$).

**Table 2.** Droplet diameters and *RS*s of adjuvant solutions.

| Adjuvant | *DV*10 (μm) | *VMD* (μm) | *DV*90 (μm) | Relative Span |
|---|---|---|---|---|
| Control | 76.1b | 151.6c | 272.4ab | 1.29ab |
| AS 100 | 68.7a | 140.9b | 264.5ab | 1.39b |
| AS A + B | 96.1c | 166.5d | 297.2b | 1.21a |
| Breakthru | 109.1d | 207.4e | 452.3c | 1.65c |
| Momentive | 104.0d | 206.6e | 479.6c | 1.82d |
| ND500 | 65.9a | 127.5a | 239.6a | 1.39b |
| ND700 | 68.1a | 130.7ab | 243.0a | 1.35ab |
| QF-LY | 94.3c | 167.4d | 297.8b | 1.22a |

Note: Data in table are the average of three replicates. Different letters (a, b, c, d and e) in the same column indicate significant differences at $p < 0.05$ level.

Compared with the control solution, the adjuvants AS A + B, Breakthru, Momentive and QF-LY can effectively increase the droplet size. The droplet spectra of AS100 and QF-LY solutions were similar, with *VMD*s of 166.5 and 167.4 μm, respectively, and the relative spans of 1.21 and 1.22, respectively. In the same way, the droplet spectra of Breakthru and Momentive solutions were similar. The *VMD*s of Breakthru and Momentive solutions were 207.4 μm and 206.6 μm, and their relative spans were 1.65 and 1.82, respectively. Therefore, the adjuvants of Breakthru and Momentive increased the droplet size, but the uniformity of the droplet size decreased.

### 3.2. Airborne and Sediment Drift Potential

Figure 3 shows the airborne drift amount of spraying liquids at each height, 2 m downwind. On the whole, the airborne drift amount went up when the vertical height was below 40 cm and went down when the vertical height was higher than 40 cm. When the heights reached 75 cm and 85 cm, the airborne drift had little difference among spraying liquids because the heights were close to that of the spraying nozzle.

Comparing the drift distribution of spraying solutions at different heights, it can be seen that the AS 100 and ND 700 solutions had more airborne drift. When the height was below 30 cm, the drift of AS 100 solution increased with increasing vertical height, and the drift amount of it was the largest among these solutions. At the heights of 45 cm and 55 cm, the drift of the ND 700 solution was higher than that of the other solutions. In contrast, the Momentive solution showed the least drift, and its drift amount was much lower than the other solutions when the height was below 60 cm.

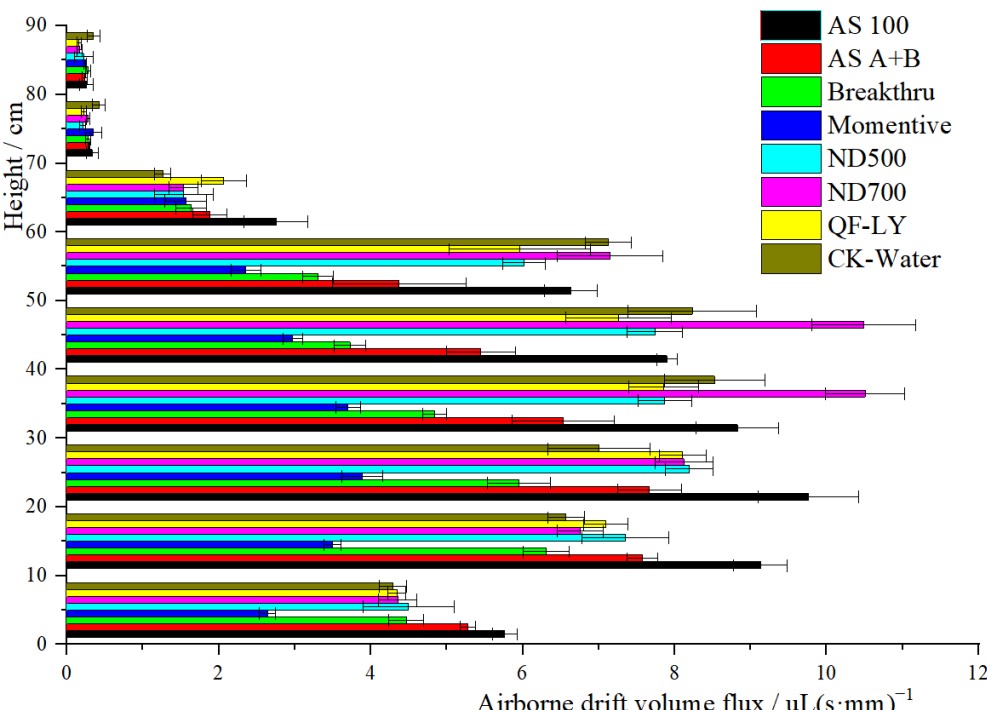

**Figure 3.** Airborne drift of adjuvant solutions in different heights.

Table 3 shows the results of airborne drift rates, characteristic heights and DIXs of spraying liquids. The airborne drift rate of the control solution was 64.33%, and the rates of ND500 and QF-LY solutions were 63.58% and 63.04%, respectively. There were no significant differences in airborne drift rates among the ND500, QF-LY and control solutions ($p < 0.05$). The airborne drift rates of the AS 100 and ND700 solutions were 74.78% and 72.72%, respectively, which were significantly greater than that of the control solution. Compared with the airborne drift rate of the control solution. The results of the AS A + B, Breakthru and Momentive solutions indicated that they could effectively reduce spraying drift, and the drift rates were 56.57%, 44.16% and 30.80%, respectively.

**Table 3.** Airborne drift results of adjuvant solutions.

| Adjuvant | Drift Rate (%) | Characteristic Height/cm | DIX (%) |
|---|---|---|---|
| Control | 64.33b | 35.77a | 100.0b |
| AS 100 | 74.78a | 34.17b | 107.9a |
| AS A + B | 56.57c | 32.78c | 83.80c |
| Breakthru | 44.16d | 32.47c | 68.48d |
| Momentive | 30.80e | 35.09ab | 55.36e |
| ND500 | 63.85b | 34.29b | 95.78b |
| ND700 | 72.72a | 35.76a | 110.0a |
| QF-LY | 63.04b | 34.68ab | 95.80b |

Note: Data in table are the average of three replicates. Different letters (a, b, c, d and e) in the same column indicate significant differences at $p < 0.05$ level.

Comparing the characteristic heights of drift of the spray liquids, the solutions of AS A + B and Breakthru had the lowest characteristic heights of 32.78 cm and 32.47 cm, followed by the AS 100 and ND 500 solutions of 34.17 cm and 34.29 cm. The characteristic height of the control solution was 35.77 cm, and the heights of the Momentive, ND 700 and QF-LY solutions were 35.09 cm, 35.76 cm and 34.68 cm, respectively. There were no significant differences in the characteristic height of the four solutions ($p < 0.05$).

The significant differences in DIXs of spraying solutions were consistent with drift rates. Compared to the DIX of the control solution (100%), the DIXs of the AS 100 and ND700 solutions were 107.9% and 110.0%, respectively, which somewhat increased the

drift potential of the liquid. The DIXs of the ND500 and QF-LY solutions were 95.78% and 95.80%, and there were no significant differences with that of the control solution. The DIXs of the AS A + B, Breakthru and Momentive solutions were 83.80%, 68.48% and 55.36%, respectively, which indicated the effectiveness of anti-drift.

Figure 4 shows the sediment drift amount of spraying liquids from 1.0 m to 3.5 m downwind. Table 4 shows the results of sediment drift rates, characteristic distances and DIXs of spraying liquids. For all the spraying solutions, the sediment drift amount decreased with increasing distance from the nozzle, and the sediment drift amounts of solutions had a similar tendency with airborne drift. The AS 100 solution showed more sediment drift at all distances downwind of the nozzle. The ND 700 solution had high airborne drift, while its sediment drift was relatively lower at 1.0 m and 1.5 m downwind of the nozzle. The Momentive solution had less sediment drift at all distances, which was exactly the same as the result of airborne drift.

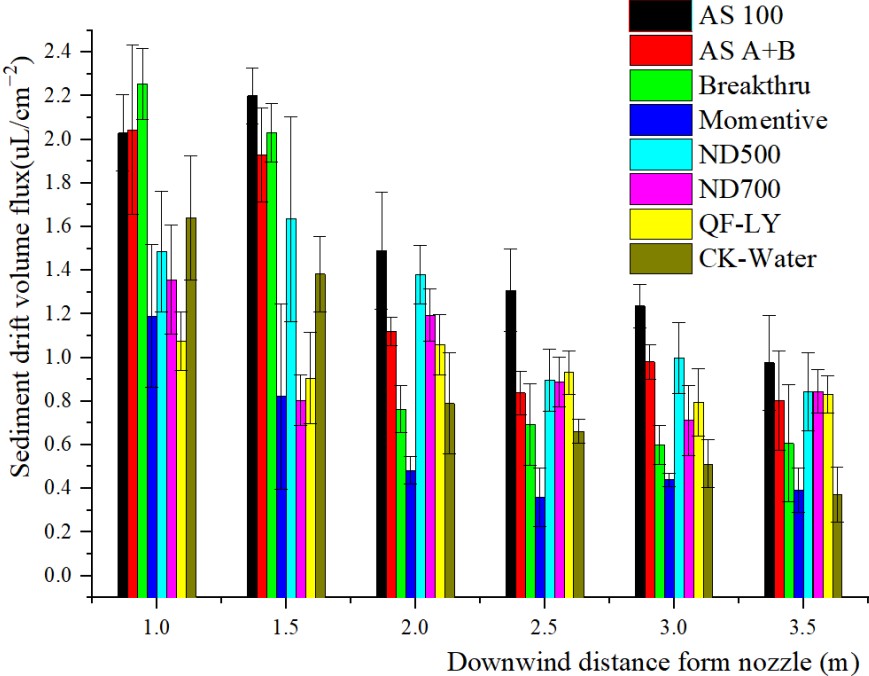

**Figure 4.** Sediment drift of adjuvant solutions in different downwind distances.

**Table 4.** Sediment drift results of adjuvant solutions.

| Adjuvant | Drift Rate (%) | Characteristic Distance/cm | DIX (%) |
|---|---|---|---|
| Control | 39.30b | 442.8cd | 100.0b |
| AS 100 | 52.08a | 455.2c | 127.2a |
| AS A + B | 42.39b | 448.3c | 107.2b |
| Breakthru | 36.60b | 422.6e | 90.8b |
| Momentive | 22.14d | 436.0d | 63.1c |
| ND500 | 41.36b | 464.6b | 108.5b |
| ND700 | 33.16c | 471.6a | 92.6b |
| QF-LY | 36.73b | 475.2a | 92.3b |

Note: Data in table are the average of three replicates. Different letters in the same column indicated significant differences at $p < 0.05$ level.

### 3.3. Field Deposition

Table 5 shows the results of average deposition, standard deviation and coefficient of variation of spraying solutions deposited in the wheat field. For the control spraying solution, the average deposition was 0.151 μL/cm$^2$, the standard deviation was 0.066 μL/cm$^2$

and the coefficient of variation of the deposited liquid was 43.84%. Compared with the deposition result of the control solution, the average deposition volumes of adjuvant solutions were between 0.140 μL/cm$^2$ and 0.160 μL/cm$^2$, and all their coefficients of variation were greater than 30%. There were no significant differences ($p < 0.05$) in the droplet deposition and coefficient of variation for all treatments.

**Table 5.** Influence of adjuvants on spray liquid deposition.

| Adjuvant | Average Deposition (μL/cm$^2$) | Standard Deviation | Coefficient of Variation/% |
|---|---|---|---|
| Control | 0.151 | 0.066 | 43.84 |
| AS 100 | 0.140 | 0.063 | 44.65 |
| AS A + B | 0.143 | 0.046 | 32.33 |
| Breakthru | 0.157 | 0.069 | 43.94 |
| Momentive | 0.160 | 0.065 | 40.64 |
| ND500 | 0.153 | 0.059 | 38.56 |
| ND700 | 0.143 | 0.056 | 39.04 |
| QF-LY | 0.159 | 0.058 | 36.56 |

*3.4. Control Efficacy*

Table 6 shows the control efficacy of spraying solutions on wheat rust 7 and 14 days after pesticides application. The control efficacies of the control solution were 44.76% and 23.95% after 7 and 14 days of application, respectively. When the adjuvants were added to the spray solutions, all the adjuvant solutions had a synergistic effect on the control of wheat rust. After 7 days of aerial application in the wheat field, the control efficacies of Momentive and ND500 solutions were 59.87% and 56.81%, respectively, and the control efficacies of other adjuvant treatments were greater than 60%. After 14 days of application, there were no significant differences in control efficacies between these adjuvant solutions. The control efficacies of adjuvant solutions were between 31.32% and 38.01%, which were significantly greater than that of the control solution.

**Table 6.** Effect of adjuvants on control efficacy of wheat rust (DAT = days after treatment).

| Adjuvants | 7 DAT | 14 DAT |
|---|---|---|
| Control | 44.76c | 23.95b |
| AS 100 | 62.26a | 35.10a |
| AS A + B | 65.99a | 35.60a |
| Breakthru | 64.41a | 35.61a |
| Momentive | 59.87ab | 31.32a |
| ND500 | 64.37a | 38.01a |
| ND700 | 56.81b | 34.72a |
| QF-LY | 69.65a | 32.20a |

Note: Data are the average of five replicates. Different letters in the same column indicate significant differences at $p < 0.05$.

Table 7 shows the control efficacies of wheat aphids of each spraying solution. Similar to the results of rust, all the adjuvant solutions can increase the control effect of pesticides on wheat aphids. After 1 day of pesticide application, the efficacy of the control solution was only 26.47%, while the efficacies of all adjuvant solutions were greater than 40%. In particular, the efficacies of the ND500, AS A + B and Breakthru solutions ranged from 52.31% to 60.04%, which were significantly better than the other four adjuvants. For the same spraying solution, the efficacy at 3 days after application was greater than that at 1 day after application. At 3 days after application, the efficacy of the control solution was 45.06%, and the efficacies of adjuvant solutions were greater than 50%.

After 7 days of pesticides application, the ND700 solution had the best efficacy at 81.77%, followed by the ND500 solution with 68.08%, and the efficacies of the other adjuvant solutions ranged from 51.55% to 58.54%. However, the efficacy of the control solution

was only 40.77% at 7 days after treatment. After 14 days of application, the efficacy of each treatment on wheat aphids was reduced. The control solution had the lowest control efficacy of 25.44%. Nevertheless, the efficacy of the ND500 solution was still as high as 63.46%, which was significantly greater than that of the other treatments ($p < 0.05$).

**Table 7.** Effect of adjuvants on control efficacy of aphid.

| Adjuvants | 1 DAT | 3 DAT | 7 DAT | 14 DAT |
|---|---|---|---|---|
| Control | 26.74c | 45.06c | 40.77d | 25.44d |
| AS 100 | 45.84b | 54.71b | 54.09c | 50.06b |
| AS A + B | 53.44a | 66.99ab | 58.54c | 53.64b |
| Breakthru | 52.31a | 75.36a | 54.13c | 47.05bc |
| Momentive | 40.23b | 54.06b | 57.57c | 40.23c |
| ND500 | 60.04a | 68.00ab | 68.08b | 63.46a |
| ND700 | 42.96b | 64.10ab | 81.17a | 45.02c |
| QF-LY | 42.22b | 51.80b | 51.55c | 32.37cd |

Note: Data are the average of five replicates. Different letters in the same column indicate significant differences at $p < 0.05$.

## 4. Discussion

Pesticide spray drift is a significant environmental problem caused by the aerial application of UAV and is a vital factor affecting the utilization and control effect of pesticides [26,27]. Spray drift was influenced by the physical properties of spray liquids, droplet size distribution, meteorological factors, operating skills and parameters [12,28–33]. However, when the operating parameters of UAVs were constant, the meteorological conditions were uncertain in practical field applications. The addition of adjuvants is the most effective method to reduce droplet drift and increase control efficacy. Adding adjuvants into the spray solution can change its physicochemical properties, thereby affecting the result of atomization, such as diameter and relative droplet span.

The amount of spray drift is usually related to the percentage of fine spray droplets [31,34,35]. The smaller a spray droplet, the longer it remains airborne and the higher the possibility for it to be carried away by crosswind [36]. In this study, the methylated vegetable oil adjuvant (AS100) and the hyperbranched polymer adjuvants (ND500 and ND700) reduced the droplet size, and their corresponding airborne drift potentials were larger. While adjuvants AS A + B, Breakthru, Momentive and QF-LY increased the droplet size, and their wind tunnel drift results also showed good performance on drift reduction. Numerous studies have shown that adjuvants can alter the size and distribution of particles during liquid atomization and influence the drift potentials of the spraying liquids [37–40]. This result once again proved that there is a strong negative correlation between the droplet size and the drift potential.

It has been recognized for decades that inconsistent spray coverages of pesticide applications represent a major challenge to successful and sustainable crop protection [41]. The addition of adjuvants into spray solution can also change the uniformity of droplet diameter, and a narrow droplet size spectrum is therefore required for optimized application. Among the adjuvants tested, the Breakthru and Momentive increased the *RS* of the droplet. This may not be conducive to the efficacy of the pesticide solution. In addition, adding adjuvants to change the droplet spectrum and increase the droplet size is beneficial to reduce drift, but a larger droplet size may reduce the control effect of pesticides [34]. However, it has also been shown that coarse droplets and narrow *RS* were not sufficient to improve the control efficacy [42,43]. The control efficacy of pesticide solutions is mainly influenced by their wetting properties and mechanism of action [34].

In the testing process of the field pesticide application, the wind speed ranged from 1.36 m/s to 2.73 m/s. The relative humidity was from 41.4% to 54.7%, and the temperature was between 28.5 °C and 30.9 °C. Because the wind speed is unstable and uncontrollable in the field [20], the wind tunnel drift reduction effect of the adjuvants was not reflected in the increase of deposition on field targets. On the whole, these adjuvants with good

drift reduction performance in the wind tunnel led to a relatively larger numeric of field deposition. In terms of the uniformity of droplet deposition, the CV of droplet deposition was higher than 30% for all treatments of aerial application, which were much higher than that of 10% sprayed with ground boom sprayers according to the machinery industry standard of China (JB/T 13854-2020: Self-propelled boom sprayers). It is worth mentioning that the application volume of the UAV was much lower than that of the boom sprayer. This indicates that besides drift reduction, the distribution uniformity of the droplets should also be considered for aerial application, such as optimizing the spray system or operating the path planning.

Currently, more and more tank-mixed adjuvants are used in aerial applications in China, mainly methyl esterified vegetable oils, organic silicon and high molecular polymers. MSO is a kind of fatty acid from seed oil esterified with methyl alcohol, which is now commonly used to enhance the control efficacy of pesticides. Xu et al. [44,45] reported that MSO could decrease the surface tension and contact angle and then increase the wetted areas of droplets on leaves. In herbicide application, some reports have shown that MSO enhances the efficacy of several herbicides on certain weed species by increasing the absorption of the herbicides by weeds [46–51]. Silicone and polymer adjuvants are mainly used to change the physical and chemical properties of the spray solution, such as surface tension, shear extensional viscosity or presence of inhomogeneities in the spray liquid [34,36]. These properties of spray liquid could greatly affect the spray angle, droplet spectrum and dynamic spreading of droplets on the targets [52–57]. Therefore, it is suggested to consider the compounding of multiple components in the development of adjuvants to optimize their performance.

The mechanism of action of adjuvants is difficult to explain, but is mainly summarized as reducing the surface tension of spray solution, increasing wetting and spreading and increasing penetration and deposition on the target [58,59]. As suggested by the comprehensive comparison of the control effects of adjuvant solutions on wheat rust and aphids at different periods after application, the adjuvants could significantly optimize the control efficacy of plant protection UAVs. This result is in accordance with those reported by Meng et al. [21] on wheat aphids.

In addition, adjuvants are beneficial to prolong the duration of pesticides. It is suggested that tank-mix adjuvants should be added when UAVs are used for aerial application. The effectiveness of pesticides are affected by many factors [34], such as pesticide deposition, meteorological conditions and the spreading and absorption of droplets on the target. Therefore, the synergistic effect of adjuvants is different from the results of the wind tunnel drift test and field deposition. The mechanisms of different adjuvants still need to be further studied.

## 5. Conclusions

The performances of various types of tank-mix adjuvants were compared by analyzing droplet spectrum, drift potential, deposition and control efficacy for aerial application. The results showed that the addition of adjuvants could change the droplet spectrum of atomization, and the droplet size is an effective indicator of spray drift potential. In the field application, the meteorological conditions are complex and uncontrollable, as a result, the anti-drift effect of the adjuvants in the wind tunnel experiment was not reflected in the field deposition. The effects of adjuvants on droplet deposition and distribution were not significant, and the CVs of deposition were higher than 30% for all treatments. Adding adjuvants to the spray solution can significantly improve the control efficacy of pesticides on wheat aphids and rust and can also prolong the duration of pesticides. Our results suggest that tank-mix adjuvants should be added when UAVs are used for aerial application. It is worth mentioning that the mechanism of action of adjuvants is difficult to explain, and the mechanisms of different adjuvants still need to be further studied.

**Author Contributions:** Conceptualization, X.H.; data curation, S.W. and X.L. (Xue Li); formal analysis, S.W. and T.X.; investigation, S.W., A.Z., J.S. and T.X.; methodology, S.W., A.Z., J.S. and X.H.; supervision, X.H.; writing—original draft, S.W. and X.L. (Xue Li); writing—review and editing, X.L. (Xue Li) and X.L. (Xiaolan Lv). All authors have read and agreed to the published version of the manuscript.

**Funding:** This research was funded by the National Natural Science Foundation of China (No. 32001954), the Natural Science Foundation of Jiangsu Province (BK20200280) and the China Postdoctoral Science Foundation (2020M671390).

**Data Availability Statement:** The data presented in this study are available on request from the corresponding author.

**Acknowledgments:** The authors would like to give special thanks to the staff Deng Xijun and Liang Zijing from Anyang Quanfeng Biological Technology Co., Ltd. for their contribution in the wind tunnel and field experiments.

**Conflicts of Interest:** The authors declare no conflict of interest.

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
