# Peer review of "Effects of Adjuvants on Spraying Characteristics and Control Efficacy in Unmanned Aerial Application"

_agriculture, doi:10.3390/agriculture12020138_

Round 1

Reviewer 1 Report

General comments:

The paper is well structured, the context well described and the scientific design is appropriate, focusing both on laboratory and field trials. The item is relevant for the perspectives of spray application using UAV that, besides Eastern Asia, are spreading up in the whole world. Methods used are well described and in line with ISO Standard 22856 for what concerns spray drift assessment in wind tunnel. Results are clearly presented and accompanied by adequate tables and figures. Discussion of results is well developed and critically conducted. Conclusions are consequent. Bibliography is relevant and pretty complete.

Specific comments:

In paragraph 2.3.3 (field tests) at line 180 it is described the composition of the spray mixtures applied in the field, mentioning the addition of the BSF tracer. Nothing is said however if preliminary tests were conducted in order to check for eventual interference on tracer fluorescence (e.g instability of fluorescence) due to pesticides and/or adjuvants. Please clarify.

In paragraph 3.2 (airborne and sediment drift potential) in Figure 3 the bars of the graph should be thicker (like the histograms in Figure 4) in order to better identify the values referred to each adjuvant.

In paragraph 4 (Discussion) at lines 383-387 it is mentioned that the CV of droplets deposition resulted always higher than 30 % for aerial application, much more than the average CV of 15% referred to ground boom sprayers. First of all it should be cited the reference for such 15% CV value, and then it should be considered that while all the aerial applications carried out in the experiments with UAV were made applying a volume of 12 L/ha, the compared spray application made with a conventional ground boom sprayer was likely made applying a much higher volume (that should be reported). In this perspective, the comparison between the CV values from UAV and ground boom sprayers should be adequately reconsidered.         

Author Response

Dear professsor, 

Thank you for your valuable comments on the manuscript. We have revised the manuscript according to your comments. The respond to specific comments are as follows:

Point 1: In paragraph 2.3.3 (field tests) at line 180 it is described the composition of the spray mixtures applied in the field, mentioning the addition of the BSF tracer. Nothing is said however if preliminary tests were conducted in order to check for eventual interference on tracer fluorescence (e.g instability of fluorescence) due to pesticides and/or adjuvants. Please clarify.

Response 1: Pesticides and adjuvants have no interfer on the stability of fluorescent. At present, BSF has been widely used as a tracer to study the drift and deposition characteristics of pesticides. In order to make sure the stability of fluorescence, we also added the relevant requirements for the collection, storage and determination of BSF samples in the manuscript. Some of the specific relevant research papers are as follows:

  1. Rousseau J C . Development of harmonised standards on environment for new sprayers[J]. Julius-Kühn-Archiv, 2013, 55(439):234-238.
  2. Qin W C , Xue X Y , Zhou Q Q , et al. Use of RhB and BSF as fluorescent tracers for determining pesticide spray distribution[J]. Analytical methods, 2018, 10(1).
  3. Wang X N, He X K, Song J L, Wang Z C, Wang C L, Wang S L, et al. Drift potential of UAV with adjuvants in aerial applications. Int J Agric & Biol Eng, 2018; 11(5): 54–58.

Point 2: In paragraph 3.2 (airborne and sediment drift potential) in Figure 3 the bars of the graph should be thicker (like the histograms in Figure 4) in order to better identify the values referred to each adjuvant.

Response 2: The bars of figure 3 has been thickened.

Point 3: In paragraph 4 (Discussion) at lines 383-387 it is mentioned that the CV of droplets deposition resulted always higher than 30 % for aerial application, much more than the average CV of 15% referred to ground boom sprayers. First of all it should be cited the reference for such 15% CV value, and then it should be considered that while all the aerial applications carried out in the experiments with UAV were made applying a volume of 12 L/ha, the compared spray application made with a conventional ground boom sprayer was likely made applying a much higher volume (that should be reported). In this perspective, the comparison between the CV values from UAV and ground boom sprayers should be adequately reconsidered.  

Response 3: The 15% CV value of the boom sprayer was based on the national standard of China GB/T24677-1-2009. In 2020, The Ministry of Industry and Information Technology of China issued the latest industry standard (JB/T 13854-2020: Self-propelled boom sprayers), stipulating that the CV value of boom sprayer should be less than 10%. Therefore, we changed the CV value in the original manuscript to 10% and stated that it was based on the Chinese industry standard JB/T 13854-2020. Similarly, the CV value of boom sprayer in Europe is more strict, but we did not find the specific standard document.

As for the difference in spray volume between UAV aerial spray and ground boom spray.The CV represents the ratio of the standard deviation to the mean, and it is a useful statistic for comparing the degree of variation from one data series to another, even if the means (depends on theapplication volume) are drastically different from one another. Nonetheless, we added the explanation of the difference in spray volume between aerial spray and ground spray in the manuscript.

Reviewer 2 Report

Review Agriculture-1514296:  Effects of adjuvants on spraying characteristics and control efficacy in unmanned aerial application

This paper concerns the improvement of UASS efficiency with the help of adjuvants. This topic is online with the huge development of UASS to spray crops in Asia. However, the paper shows major methodological weaknesses that limits the interpretation of results.

  • Droplet size and DIX measurements were realized considering only adjuvants with water, unless other indication in the text, as biological efficacy was tested considering adjuvants + fungicide or insecticide that is not totally comparable and eventual correlations between both approaches are not possible.  
  • It would be recommended to complete the experimental setup for droplet size and DIX with adjuvants and chemicals or to limit the scope of the paper to data concerning water + adjuvants only.

  • DIX measurements considered a wind tunnel with a quite ow cross section of 1m x 1m that is 2 times lower than the recommendation of ISO 22856. The use of large spray angle such as 120° at a height of 0.85m involves that the spray footprint on the ground is higher than 2m with potential consequence of spraying on sidewalls of this narrow wind tunnel. This probably limits the interpretation of results.

  • Biological efficacy protocols are not explained in the material and method section: is there any control to assess the desease/pest severity without application? As I understand, the control is the spray with chemical but without adjuvants.

Author Response

Dear professor, 

Thank you for your valuable comments on the manuscript. We have revised the manuscript according to your comments. The respond to specific comments are as follows:

Point 1: Droplet size and DIX measurements were realized considering only adjuvants with water, unless other indication in the text, as biological efficacy was tested considering adjuvants + fungicide or insecticide that is not totally comparable and eventual correlations between both approaches are not possible.  

It would be recommended to complete the experimental setup for droplet size and DIX with adjuvants and chemicals or to limit the scope of the paper to data concerning water + adjuvants only.

Response 1: Pesticides are a complex system, and their types, formulations, manufacturers, and concentrations all have a significant influence on the properties of the solution, and the influence  trend cannot be judged. In this study, we mainly compared the effects of adjuvants on the effect of drones. If a particular pesticide is added to the measurement, the application and guidance of the study will be greatly reduced. Therefore, we did not add a specific pesticide to the solution when measuring DIX and droplet size. However, the ultimate goal of pesticides application is to have a good control efficacy. In this case, the 20% imidacloprid SL (ai.60 g/ha) and 40% tebuconazole SC (ai.480 g/ha) were added into the adjuvant solutions in the field test. The applied pesticides are only to verify the effect of adjuvants on the control efficacy. Of course, the effects of other pesticides on the control efficacy should be tested through more field trials. Since field testing requires a lot of time and labor, we did not carry out field testing on more pesticides. We sincerely hope that you can accept our explanation.

Point 2: DIX measurements considered a wind tunnel with a quite ow cross section of 1m x 1m that is 2 times lower than the recommendation of ISO 22856. The use of large spray angle such as 120° at a height of 0.85m involves that the spray footprint on the ground is higher than 2m with potential consequence of spraying on sidewalls of this narrow wind tunnel. This probably limits the interpretation of results.

Response 2: Your comment is correct. The ISO 22856 specified wind tunnel cross section is 1m(height) x 2m (windth). The ISO standard wind tunnel was too expensive for us. Except the windth of wind tunnel, the height of the wind tunnel, the length of the working section, and the air flow parameters all meet the requirements. Our testing methods are also strictly in accordance with ISO standards. Although the spray angle of the nozzle is 120°, the speed of the droplets is rapidly attenuated after spraying from the jet orifice, so the moving distance of the droplets in the lateral direction is very limited, and the coverage distance in the windth range is no more than 1m. Taking the boom sprayer as an example, the installation distance of the nozzle on the boom is about 0.4-0.5m. Fortunately, in the actual wind tunnel test, there were no droplets spraying on sidewalls of the wind tunnel, which had no effect on the test results.

Point 3: Biological efficacy protocols are not explained in the material and method section: is there any control to assess the desease/pest severity without application? As I understand, the control is the spray with chemical but without adjuvants.

Response 3: We have supplemented the biological efficacy protocols of wheat aphid and rust (2.4.3. Control efficacy of pesticides). Due to the previous version did not introduce the calculation method of the control efficacy, the untreated plot was not explained in the original text. Actually, there were 9 plots, of which 7 plots were sprayed with adjuvants added, one plot was sprayed without adjuvants as a control, and another plot was to assess the disease and pest severity without application. The untreated plot was to assess the desease/pest severity. These have been modified and explained in the manuscript.

Reviewer 3 Report

The article presents an important experience on the use of adjuvants in crop protection technology. The research is conducted well and the results are presented appropriately. Research on spray drift has been carried out for a long time in many scientific centers. The implementation of adjuvants into agricultural technology forces the search for new environmentally safe solutions. 

Author Response

Dear professor, 
Thank you for your valuable comments on the manuscript. We have revised the manuscript according to your comments. Please refer to the red font part of the manuscript for the details of the revision.

Round 2

Reviewer 2 Report

The text was significantly improved.